# Accuracy First: Selecting a Differential Privacy Level for Accuracy-Constrained ERM

**Katrina Ligett**
Caltech and Hebrew University

**Seth Neel**
University of Pennsylvania

**Aaron Roth**
University of Pennsylvania

**Bo Waggoner**
University of Pennsylvania

**Zhiwei Steven Wu**
Microsoft Research

## Abstract

Traditional approaches to differential privacy assume a fixed privacy requirement $\varepsilon$ for a computation, and attempt to maximize the accuracy of the computation subject to the privacy constraint. As differential privacy is increasingly deployed in practical settings, it may often be that there is instead a fixed accuracy requirement for a given computation and the data analyst would like to maximize the privacy of the computation subject to the accuracy constraint. This raises the question of how to find and run a maximally private empirical risk minimizer subject to a given accuracy requirement. We propose a general "noise reduction" framework that can apply to a variety of private empirical risk minimization (ERM) algorithms, using them to "search" the space of privacy levels to find the empirically strongest one that meets the accuracy constraint, and incurring only logarithmic overhead in the number of privacy levels searched. The privacy analysis of our algorithm leads naturally to a version of differential privacy where the privacy parameters are dependent on the data, which we term *ex-post* privacy, and which is related to the recently introduced notion of privacy odometers. We also give an *ex-post* privacy analysis of the classical AboveThreshold privacy tool, modifying it to allow for queries chosen depending on the database. Finally, we apply our approach to two common objective functions, regularized linear and logistic regression, and empirically compare our noise reduction methods to (i) inverting the theoretical utility guarantees of standard private ERM algorithms and (ii) a stronger, empirical baseline based on binary search.[1]

## 1 Introduction and Related Work

Differential Privacy [7, 8] enjoys over a decade of study as a theoretical construct, and a much more recent set of large-scale practical deployments, including by Google [10] and Apple [11]. As the large theoretical literature is put into practice, we start to see disconnects between assumptions implicit in the theory and the practical necessities of applications. In this paper we focus our attention on one such assumption in the domain of private empirical risk minimization (ERM): that the data analyst first chooses a privacy requirement, and then attempts to obtain the best accuracy guarantee (or empirical performance) that she can, given the chosen privacy constraint. Existing theory is tailored to this view: the data analyst can pick her privacy parameter $\varepsilon$ via some exogenous process, and either plug it into a "utility theorem" to upper bound her accuracy loss, or simply deploy her algorithm and (privately) evaluate its performance. There is a rich and substantial literature on private convex ERM that takes this approach, weaving tight connections between standard mechanisms in

differential privacy and standard tools for empirical risk minimization. These methods for private ERM include output and objective perturbation [5, 14, 18, 4], covariance perturbation [19], the exponential mechanism [16, 2], and stochastic gradient descent [2, 21, 12, 6, 20].

While these existing algorithms take a privacy-first perspective, in practice, product requirements may impose hard accuracy constraints, and privacy (while desirable) may not be the over-riding concern. In such situations, things are reversed: the data analyst first fixes an accuracy requirement, and then would like to find the smallest privacy parameter consistent with the accuracy constraint. Here, we find a gap between theory and practice. The only theoretically sound method available is to take a "utility theorem" for an existing private ERM algorithm and solve for the smallest value of $\varepsilon$ (the differential privacy parameter)—and other parameter values that need to be set—consistent with her accuracy requirement, and then run the private ERM algorithm with the resulting $\varepsilon$. But because utility theorems tend to be worst-case bounds, this approach will generally be extremely conservative, leading to a much larger value of $\varepsilon$ (and hence a much larger leakage of information) than is necessary for the problem at hand. Alternately, the analyst could attempt an empirical search for the smallest value of $\varepsilon$ consistent with her accuracy goals. However, because this search is itself a data-dependent computation, it incurs the overhead of additional privacy loss. Furthermore, it is not *a priori* clear how to undertake such a search with nontrivial privacy guarantees for two reasons: first, the worst case could involve a very long search which reveals a large amount of information, and second, the selected privacy parameter is now itself a data-dependent quantity, and so it is not sensible to claim a "standard" guarantee of differential privacy for any finite value of $\varepsilon$ ex-ante.

In this paper, we provide a principled variant of this second approach, which attempts to empirically find the smallest value of $\varepsilon$ consistent with an accuracy requirement. We give a meta-method that can be applied to several interesting classes of private learning algorithms and introduces very little privacy overhead as a result of the privacy-parameter search. Conceptually, our meta-method initially computes a very private hypothesis, and then gradually subtracts noise (making the computation less and less private) until a sufficient level of accuracy is achieved. One key technique that significantly reduces privacy loss over naive search is the use of correlated noise generated by the method of [15], which formalizes the conceptual idea of "subtracting" noise without incurring additional privacy overhead. In order to select the most private of these queries that meets the accuracy requirement, we introduce a natural modification of the now-classic AboveThreshold algorithm [8], which iteratively checks a sequence of queries on a dataset and privately releases the index of the first to approximately exceed some fixed threshold. Its privacy cost increases only logarithmically with the number of queries. We provide an analysis of AboveThreshold that holds even if the queries themselves are the result of differentially private computations, showing that if AboveThreshold terminates after $t$ queries, one only pays the privacy costs of AboveThreshold plus the privacy cost of revealing those first $t$ private queries. When combined with the above-mentioned correlated noise technique of [15], this gives an algorithm whose privacy loss is *equal* to that of the final hypothesis output – the previous ones coming "for free" – plus the privacy loss of AboveThreshold. Because the privacy guarantees achieved by this approach are not fixed a priori, but rather are a function of the data, we introduce and apply a new, corresponding privacy notion, which we term *ex-post* privacy, and which is closely related to the recently introduced notion of "privacy odometers" [17].

In Section 4, we empirically evaluate our noise reduction meta-method, which applies to any ERM technique which can be described as a post-processing of the Laplace mechanism. This includes both direct applications of the Laplace mechanism, like *output perturbation* [5]; and more sophisticated methods like *covariance perturbation* [19], which perturbs the covariance matrix of the data and then performs an optimization using the noisy data. Our experiments concentrate on $\ell_2$ regularized least-squares regression and $\ell_2$ regularized logistic regression, and we apply our noise reduction meta-method to both output perturbation and covariance perturbation. Our empirical results show that the active, ex-post privacy approach massively outperforms inverting the theory curve, and also improves on a baseline "$\varepsilon$-doubling" approach.

## 2 Privacy Background and Tools

### 2.1 Differential Privacy and Ex-Post Privacy

Let $\mathcal{X}$ denote the data domain. We call two *datasets* $D, D' \in \mathcal{X}^*$ *neighbors* (written as $D \sim D'$) if $D$ can be derived from $D'$ by replacing a single data point with some other element of $\mathcal{X}$.

**Definition 2.1** (Differential Privacy [7])**.** Fix $\varepsilon \geq 0$. A randomized algorithm $A : \mathcal{X}^* \rightarrow \mathcal{O}$ is $\varepsilon$-differentially private if for every pair of neighboring data sets $D \sim D' \in \mathcal{X}^*$, and for every event $S \subseteq \mathcal{O}$:

$$\Pr[A(D) \in S] \leq \exp(\varepsilon) \Pr[A(D') \in S].$$

We call $\exp(\varepsilon)$ the *privacy risk* factor.

It is possible to design computations that do not satisfy the differential privacy definition, but whose outputs are private to an extent that can be quantified after the computation halts. For example, consider an experiment that repeatedly runs an $\varepsilon'$-differentially private algorithm, until a stopping condition defined by the output of the algorithm itself is met. This experiment does not satisfy $\varepsilon$-differential privacy for any fixed value of $\varepsilon$, since there is no fixed maximum number of rounds for which the experiment will run (for a fixed number of rounds, a simple composition theorem, Theorem 2.5, shows that the $\varepsilon$-guarantees in a sequence of computations "add up.") However, if ex-post we see that the experiment has stopped after $k$ rounds, the data can in some sense be assured an "ex-post privacy loss" of only $k\varepsilon'$. Rogers et al. [17] initiated the study of *privacy odometers*, which formalize this idea. They study privacy composition when the data analyst can choose the privacy parameters of subsequent computations as a function of the outcomes of previous computations.

We apply a related idea here, for a different purpose. Our goal is to design one-shot algorithms that always achieve a target accuracy but that may have variable privacy levels depending on their input.

**Definition 2.2.** Given a randomized algorithm $\mathcal{A} : \mathcal{X}^* \rightarrow \mathcal{O}$, define the *ex-post privacy loss*[2] of $\mathcal{A}$ on outcome $o$ to be

$$\text{Loss}(o) = \max_{D, D':D \sim D'} \log \frac{\Pr\left[\mathcal{A}(D) = o\right]}{\Pr\left[\mathcal{A}(D') = o\right]}.$$

We refer to $\exp\left(\text{Loss}(o)\right)$ as the *ex-post privacy risk* factor.

**Definition 2.3** (Ex-Post Differential Privacy)**.** Let $\mathcal{E} : \mathcal{O} \rightarrow (\mathbb{R}_{\geq 0} \cup \{\infty\})$ be a function on the outcome space of algorithm $\mathcal{A} : \mathcal{X}^* \rightarrow \mathcal{O}$. Given an outcome $o = A(D)$, we say that $\mathcal{A}$ satisfies $\mathcal{E}(o)$-*ex-post* differential privacy if for all $o \in \mathcal{O}$, $\text{Loss}(o) \leq \mathcal{E}(o)$.

Note that if $\mathcal{E}(o) \leq \varepsilon$ for all $o$, $\mathcal{A}$ is $\varepsilon$-differentially private. Ex-post differential privacy has the same semantics as differential privacy, once the output of the mechanism is known: it bounds the log-likelihood ratio of the dataset being $D$ vs. $D'$, which controls how an adversary with an arbitrary prior on the two cases can update her posterior.

## 2.2 Differential Privacy Tools

Differentially private computations enjoy two nice properties:

**Theorem 2.4** (Post Processing [7])**.** *Let $A : \mathcal{X}^* \rightarrow \mathcal{O}$ be any $\varepsilon$-differentially private algorithm, and let $f : \mathcal{O} \rightarrow \mathcal{O}'$ be any function. Then the algorithm $f \circ A : \mathcal{X}^* \rightarrow \mathcal{O}'$ is also $\varepsilon$-differentially private.*

Post-processing implies that, for example, every *decision* process based on the output of a differentially private algorithm is also differentially private.

**Theorem 2.5** (Composition [7])**.** *Let $A_1 : \mathcal{X}^* \rightarrow \mathcal{O}$, $A_2 : \mathcal{X}^* \rightarrow \mathcal{O}'$ be algorithms that are $\varepsilon_1$- and $\varepsilon_2$-differentially private, respectively. Then the algorithm $A : \mathcal{X}^* \rightarrow \mathcal{O} \times \mathcal{O}'$ defined as $A(x) = (A_1(x), A_2(x))$ is $(\varepsilon_1 + \varepsilon_2)$-differentially private.*

The composition theorem holds even if the composition is *adaptive*—-see [9] for details.

**The Laplace mechanism.** The most basic subroutine we will use is the *Laplace mechanism*. The Laplace Distribution centered at $0$ with scale $b$ is the distribution with probability density function $\text{Lap}\left(z|b\right) = \frac{1}{2b}e^{-\frac{|z|}{b}}$. We say $X \sim \text{Lap}\left(b\right)$ when $X$ has Laplace distribution with scale $b$. Let $f \colon \mathcal{X}^* \rightarrow \mathbb{R}^d$ be an arbitrary $d$-dimensional function. The $\ell_1$ *sensitivity* of $f$ is defined to be $\Delta_1(f) = \max_{D \sim D'} \|f(D) - f(D')\|_1$. The *Laplace mechanism* with parameter $\varepsilon$ simply adds noise drawn independently from $\text{Lap}\left(\frac{\Delta_1(f)}{\varepsilon}\right)$ to each coordinate of $f(x)$.

**Theorem 2.6** ([7]). *The Laplace mechanism is $\varepsilon$-differentially private.*

**Gradual private release.** Koufogiannis et al. [15] study how to gradually release private data using the Laplace mechanism with an increasing sequence of $\varepsilon$ values, with a privacy cost scaling only with the privacy of the *marginal* distribution on the least private release, rather than the sum of the privacy costs of independent releases. For intuition, the algorithm can be pictured as a continuous random walk starting at some private data $v$ with the property that the marginal distribution at each point in time is Laplace centered at $v$, with variance increasing over time. Releasing the value of the random walk at a fixed point in time gives a certain output distribution, for example, $\hat{v}$, with a certain privacy guarantee $\varepsilon$. To produce $\hat{v}'$ whose *ex-ante* distribution has higher variance (is more private), one can simply "fast forward" the random walk from a starting point of $\hat{v}$ to reach $\hat{v}'$; to produce a less private $\hat{v}'$, one can "rewind." The total privacy cost is $\max\{\varepsilon, \varepsilon'\}$ because, given the "least private" point (say $\hat{v}$), all "more private" points can be derived as post-processings given by taking a random walk of a certain length starting at $\hat{v}$. Note that were the Laplace random variables used for each release independent, the composition theorem would require *summing* the $\varepsilon$ values of all releases.

In our private algorithms, we will use their noise reduction mechanism as a building block to generate a list of private hypotheses $\theta^1, \ldots, \theta^T$ with gradually increasing $\varepsilon$ values. Importantly, releasing any prefix $(\theta^1, \ldots, \theta^t)$ only incurs the privacy loss in $\theta^t$. More formally:

---

**Algorithm 1** Noise Reduction [15]: $\mathrm{NR}(v, \Delta, \{\varepsilon_t\})$

---

**Input:** private vector $v$, sensitivity parameter $\Delta$, list $\varepsilon_1 < \varepsilon_2 < \cdots < \varepsilon_T$
Set $\hat{v}_T := v + \mathrm{Lap}\left(\Delta/\varepsilon_T\right)$                            $\triangleright$ drawn i.i.d. for each coordinate
**for** $t = T-1, T-2, \ldots, 1$ **do**
    With probability $\left(\frac{\varepsilon_t}{\varepsilon_{t+1}}\right)^2$: set $\hat{v}_t := \hat{v}_{t+1}$
    Else: set $\hat{v}_t := \hat{v}_{t+1} + \mathrm{Lap}\left(\Delta/\varepsilon_t\right)$             $\triangleright$ drawn i.i.d. for each coordinate
Return $\hat{v}_1, \ldots, \hat{v}_T$

---

**Theorem 2.7** ([15]). *Let $f$ have $\ell_1$ sensitivity $\Delta$ and let $\hat{v}_1, \ldots, \hat{v}_T$ be the output of Algorithm 1 on $v = f(D)$, $\Delta$, and the increasing list $\varepsilon_1, \ldots, \varepsilon_T$. Then for any $t$, the algorithm which outputs the prefix $(\hat{v}_1, \ldots, \hat{v}_t)$ is $\varepsilon_t$-differentially private.*

### 2.3 AboveThreshold with Private Queries

Our high-level approach to our eventual ERM problem will be as follows: Generate a sequence of hypotheses $\theta_1, \ldots, \theta_T$, each with increasing accuracy and decreasing privacy; then test their accuracy levels sequentially, outputting the first one whose accuracy is "good enough." The classical AboveThreshold algorithm [8] takes in a dataset and a sequence of queries and privately outputs the index of the first query to exceed a given threshold (with some error due to noise). We would like to use AboveThreshold to perform these accuracy checks, but there is an important obstacle: for us, the "queries" themselves depend on the private data.[3] A standard composition analysis would involve first privately publishing *all* the queries, then running AboveThreshold on these queries (which are now public). Intuitively, though, it would be much better to generate and publish the queries one at a time, until AboveThreshold halts, at which point one would not publish any more queries. The problem with analyzing this approach is that, a-priori, we do not know when AboveThreshold will terminate; to address this, we analyze the *ex-post privacy* guarantee of the algorithm.[4]

Let us say that an algorithm $M(D) = (f_1, \ldots, f_T)$ is $(\varepsilon_1, \ldots, \varepsilon_T)$-*prefix-private* if for each $t$, the function that runs $M(D)$ and outputs just the prefix $(f_1, \ldots, f_t)$ is $\varepsilon_t$-differentially private.

**Lemma 2.8.** *Let $M : \mathcal{X}^* \to (\mathcal{X}^* \to \mathcal{O})^T$ be a $(\varepsilon_1, \ldots, \varepsilon_T)$-prefix private algorithm that returns $T$ queries, and let each query output by $M$ have $\ell_1$ sensitivity at most $\Delta$. Then Algorithm 2 run on $D$, $\varepsilon_A$, $W$, $\Delta$, and $M$ is $\mathcal{E}$-ex-post differentially private for $\mathcal{E}((t, \cdot)) = \varepsilon_A + \varepsilon_t$ for any $t \in [T]$.*

**Algorithm 2** InteractiveAboveThreshold: $\text{IAT}(D, \varepsilon, W, \Delta, M)$

---

**Input:** Dataset $D$, privacy loss $\varepsilon$, threshold $W$, $\ell_1$ sensitivity $\Delta$, algorithm $M$

Let $\hat{W} = W + \text{Lap}\left(\frac{2\Delta}{\varepsilon}\right)$

**for** each query $t = 1, \ldots, T$ **do**

    Query $f_t \leftarrow M(D)_t$

    **if** $f_t(D) + \text{Lap}\left(\frac{4\Delta}{\varepsilon}\right) \geq \hat{W}$: **then** Output $(t, f_t)$; **Halt.**

Output $(T, \perp)$.

---

The proof, which is a variant on the proof of privacy for AboveThreshold [8], appears in the full version, along with an accuracy theorem for IAT.

## 3 Noise-Reduction with Private ERM

In this section, we provide a general private ERM framework that allows us to approach the best privacy guarantee achievable on the data given a target excess risk goal. Throughout the section, we consider an input dataset $D$ that consists of $n$ row vectors $X_1, X_2, \ldots, X_n \in \mathbb{R}^p$ and a column $y \in \mathbb{R}^n$. We will assume that each $\|X_i\|_1 \leq 1$ and $|y_i| \leq 1$. Let $d_i = (X_i, y_i) \in \mathbb{R}^{p+1}$ be the $i$-th data record. Let $\ell$ be a loss function such that for any hypothesis $\theta$ and any data point $(X_i, y_i)$ the loss is $\ell(\theta, (X_i, y_i))$. Given an input dataset $D$ and a regularization parameter $\lambda$, the goal is to minimize the following regularized empirical loss function over some feasible set $C$:

$$L(\theta, D) = \frac{1}{n}\sum_{i=1}^{n} \ell(\theta, (X_i, y_i)) + \frac{\lambda}{2}\|\theta\|_2^2.$$

Let $\theta^* = \text{argmin}_{\theta \in C} \ell(\theta, D)$. Given a target accuracy parameter $\alpha$, we wish to privately compute a $\theta_p$ that satisfies $L(\theta_p, D) \leq L(\theta^*, D) + \alpha$, while achieving the best ex-post privacy guarantee. For simplicity, we will sometimes write $L(\theta)$ for $L(\theta, D)$.

One simple baseline approach is a "doubling method": Start with a small $\varepsilon$ value, run an $\varepsilon$-differentially private algorithm to compute a hypothesis $\theta$ and use the Laplace mechanism to estimate the excess risk of $\theta$; if the excess risk is lower than the target, output $\theta$; otherwise double the value of $\varepsilon$ and repeat the same process. (See the full version for details.) As a result, we pay for privacy loss for every hypothesis we compute and every excess risk we estimate.

In comparison, our meta-method provides a more cost-effective way to select the privacy level. The algorithm takes a more refined set of privacy levels $\varepsilon_1 < \ldots < \varepsilon_T$ as input and generates a sequence of hypotheses $\theta^1, \ldots, \theta^T$ such that the generation of each $\theta^t$ is $\varepsilon_t$-private. Then it releases the hypotheses $\theta^t$ in order, halting as soon as a released hypothesis meets the accuracy goal. Importantly, there are two key components that reduce the privacy loss in our method:

1. We use Algorithm 1, the "noise reduction" method of [15], for generating the sequence of hypotheses: we first compute a very private and noisy $\theta^1$, and then obtain the subsequent hypotheses by gradually "de-noising" $\theta^1$. As a result, any prefix $(\theta^1, \ldots, \theta^k)$ incurs a privacy loss of only $\varepsilon_k$ (as opposed to $(\varepsilon_1 + \ldots + \varepsilon_k)$ if the hypotheses were independent).

2. When evaluating the excess risk of each hypothesis, we use Algorithm 2, Interactive-AboveThreshold, to determine if its excess risk exceeds the target threshold. This incurs substantially less privacy loss than independently evaluating the excess risk of each hypothesis using the Laplace mechanism (and hence allows us to search a finer grid of values).

For the rest of this section, we will instantiate our method concretely for two ERM problems: ridge regression and logistic regression. In particular, our noise-reduction method is based on two private ERM algorithms: the recently introduced covariance perturbation technique [19] and the output perturbation method [5].

## 3.1 Covariance Perturbation for Ridge Regression

In ridge regression, we consider the squared loss function: $\ell((X_i, y_i), \theta) = \frac{1}{2}(y_i - \langle \theta, X_i \rangle)^2$, and hence empirical loss over the data set is defined as

$$L(\theta, D) = \frac{1}{2n}\|y - X\theta\|_2^2 + \frac{\lambda\|\theta\|_2^2}{2},$$

where $X$ denotes the $(n \times p)$ matrix with row vectors $X_1, \ldots, X_n$ and $y = (y_1, \ldots, y_n)$. Since the optimal solution for the unconstrained problem has $\ell_2$ norm no more than $\sqrt{1/\lambda}$ (see the full version for a proof), we will focus on optimizing $\theta$ over the constrained set $C = \{a \in \mathbb{R}^p \mid \|a\|_2 \leq \sqrt{1/\lambda}\}$, which will be useful for bounding the $\ell_1$ sensitivity of the empirical loss.

Before we formally introduce the covariance perturbation algorithm due to [19], observe that the optimal solution $\theta^*$ can be computed as

$$\theta^* = \operatorname*{argmin}_{\theta \in C} L(\theta, D) = \operatorname*{argmin}_{\theta \in C} \frac{(\theta^\mathsf{T}(X^\mathsf{T}X)\theta - 2\langle X^\mathsf{T}y, \theta \rangle)}{2n} + \frac{\lambda\|\theta\|_2^2}{2}.$$

In other words, $\theta^*$ only depends on the private data through $X^\mathsf{T}y$ and $X^\mathsf{T}X$. To compute a private hypothesis, the covariance perturbation method simply adds Laplace noise to each entry of $X^\mathsf{T}y$ and $X^\mathsf{T}X$ (the covariance matrix), and solves the optimization based on the noisy matrix and vector. The formal description of the algorithm and its guarantee are in Theorem 3.1. Our analysis differs from the one in [19] in that their paper considers the "local privacy" setting, and also adds Gaussian noise whereas we use Laplace. The proof is deferred to the full version.

**Theorem 3.1.** *Fix any $\varepsilon > 0$. For any input data set $D$, consider the mechanism $\mathcal{M}$ that computes*

$$\theta_p = \operatorname*{argmin}_{\theta \in C} \frac{1}{2n}\left(\theta^\mathsf{T}(X^\mathsf{T}X + B)\theta - 2\langle X^\mathsf{T}y + b, \theta \rangle\right) + \frac{\lambda\|\theta\|_2^2}{2},$$

*where $B \in \mathbb{R}^{p \times p}$ and $b \in \mathbb{R}^{p \times 1}$ are random Laplace matrices such that each entry of $B$ and $b$ is drawn from $Lap(4/\varepsilon)$. Then $\mathcal{M}$ satisfies $\varepsilon$-differential privacy and the output $\theta_p$ satisfies*

$$\mathbb{E}_{B,b}\left[L(\theta_p) - L(\theta^*)\right] \leq \frac{4\sqrt{2}(2\sqrt{p/\lambda} + p/\lambda)}{n\varepsilon}.$$

In our algorithm COVNR, we will apply the noise reduction method, Algorithm 1, to produce a sequence of noisy versions of the private data $(X^\mathsf{T}X, X^\mathsf{T}y)$: $(Z^1, z^1), \ldots, (Z^T, z^T)$, one for each privacy level. Then for each $(Z^t, z^t)$, we will compute the private hypothesis by solving the noisy version of the optimization problem in Equation (1). The full description of our algorithm COVNR is in Algorithm 3, and satisfies the following guarantee:

**Theorem 3.2.** *The instantiation of $\text{COVNR}(D, \{\varepsilon_1, \ldots, \varepsilon_T\}, \alpha, \gamma)$ outputs a hypothesis $\theta_p$ that with probability $1 - \gamma$ satisfies $L(\theta_p) - L(\theta^*) \leq \alpha$. Moreover, it is $\mathcal{E}$-ex-post differentially private, where the privacy loss function $\mathcal{E}: ((([T] \cup \{\perp\}) \times \mathbb{R}^p) \to (\mathbb{R}_{\geq 0} \cup \{\infty\})$ is defined as $\mathcal{E}((k, \cdot)) = \varepsilon_0 + \varepsilon_k$ for any $k \neq \perp$, $\mathcal{E}((\perp, \cdot)) = \infty$, and*

$$\varepsilon_0 = \frac{16(\sqrt{1/\lambda} + 1)^2 \log(2T/\gamma)}{n\alpha}$$

*is the privacy loss incurred by* IAT.

## 3.2 Output Perturbation for Logistic Regression

Next, we show how to combine the output perturbation method with noise reduction for the ridge regression problem.[5] In this setting, the input data consists of $n$ labeled examples $(X_1, y_1), \ldots, (X_n, y_n)$, such that for each $i$, $X_i \in \mathbb{R}^p$, $\|X_i\|_1 \leq 1$, and $y_i \in \{-1, 1\}$. The goal is to train a linear classifier given by a weight vector $\theta$ for the examples from the two classes. We consider the logistic loss function: $\ell(\theta, (X_i, y_i)) = \log(1 + \exp(-y_i\theta^\mathsf{T}X_i))$, and the empirical loss is

$$L(\theta, D) = \frac{1}{n}\sum_{i=1}^{n} \log(1 + \exp(-y_i\theta^\mathsf{T}X_i)) + \frac{\lambda\|\theta\|_2^2}{2}.$$

**Algorithm 3** Covariance Perturbation with Noise-Reduction: $\text{CovNR}(D, \{\varepsilon_1, \ldots, \varepsilon_T\}, \alpha, \gamma)$

---

**Input:** private data set $D = (X, y)$, accuracy parameter $\alpha$, privacy levels $\varepsilon_1 < \varepsilon_2 < \ldots < \varepsilon_T$, and failure probability $\gamma$

Instantiate InteractiveAboveThreshold: $\mathcal{A} = \text{IAT}(D, \varepsilon_0, -\alpha/2, \Delta, \cdot)$ with $\varepsilon_0 = 16\Delta(\log(2T/\gamma))/\alpha$ and $\Delta = (\sqrt{1/\lambda} + 1)^2/(n)$

Let $C = \{a \in \mathbb{R}^p \mid \|a\|_2 \leq \sqrt{1/\lambda}\}$ and $\theta^* = \operatorname{argmin}_{\theta \in C} L(\theta)$

Compute noisy data:

$$\{Z^t\} = \text{NR}((X^{\mathsf{T}}X), 2, \{\varepsilon_1/2, \ldots, \varepsilon_T/2\}), \qquad \{z^t\} = \text{NR}((X^{\mathsf{T}}Y), 2, \{\varepsilon_1/2, \ldots, \varepsilon_T/2\})$$

**for** $t = 1, \ldots, T$: **do**

$$\theta^t = \operatorname*{argmin}_{\theta \in C} \frac{1}{2n} \left( \theta^{\mathsf{T}} Z^t \theta - 2\langle z^t, \theta \rangle \right) + \frac{\lambda \|\theta\|_2^2}{2} \tag{1}$$

  Let $f^t(D) = L(\theta^*, D) - L(\theta^t, D)$; Query $\mathcal{A}$ with query $f^t$ to check accuracy
  **if** $\mathcal{A}$ returns $(t, f^t)$ **then Output** $(t, \theta^t)$               ▷ Accurate hypothesis found.
**Output:** $(\perp, \theta^*)$

---

The output perturbation method simply adds Laplace noise to perturb each coordinate of the optimal solution $\theta^*$. The following is the formal guarantee of output perturbation. Our analysis deviates slightly from the one in [5] since we are adding Laplace noise (see the full version).

**Theorem 3.3.** *Fix any $\varepsilon > 0$. Let $r = \frac{2\sqrt{p}}{n\lambda\varepsilon}$. For any input dataset $D$, consider the mechanism that first computes $\theta^* = \operatorname{argmin}_{\theta \in \mathbb{R}^p} L(\theta)$, then outputs $\theta_p = \theta^* + b$, where $b$ is a random vector with its entries drawn i.i.d. from Lap $(r)$. Then $\mathcal{M}$ satisfies $\varepsilon$-differential privacy, and $\theta_p$ has excess risk*

$$\mathbb{E}_b \left[ L(\theta_p) - L(\theta^*) \right] \leq \frac{2\sqrt{2}p}{n\lambda\varepsilon} + \frac{4p^2}{n^2\lambda\varepsilon^2}.$$

Given the output perturbation method, we can simply apply the noise reduction method NR to the optimal hypothesis $\theta^*$ to generate a sequence of noisy hypotheses. We will again use Interactive-AboveThreshold to check the excess risk of the hypotheses. The full algorithm OUTPUTNR follows the same structure in Algorithm 3, and we defer the formal description to the full version.

**Theorem 3.4.** *The instantiation of OUTPUTNR$(D, \varepsilon_0, \{\varepsilon_1, \ldots, \varepsilon_T\}, \alpha, \gamma)$ is $\mathcal{E}$-ex-post differentially private and outputs a hypothesis $\theta_p$ that with probability $1 - \gamma$ satisfies $L(\theta_p) - L(\theta^*) \leq \alpha$, where the privacy loss function $\mathcal{E} \colon (([T] \cup \{\perp\}) \times \mathbb{R}^p) \to (\mathbb{R}_{\geq 0} \cup \{\infty\})$ is defined as $\mathcal{E}((k, \cdot)) = \varepsilon_0 + \varepsilon_k$ for any $k \neq \perp$, $\mathcal{E}((\perp, \cdot)) = \infty$, and*

$$\varepsilon_0 \leq \frac{32 \log(2T/\gamma) \sqrt{2 \log 2/\lambda}}{n\alpha}$$

*is the privacy loss incurred by* IAT.

*Proof sketch of Theorems 3.2 and 3.4.* The accuracy guarantees for both algorithms follow from an accuracy guarantee of the IAT algorithm (a variant on the standard AboveThreshold bound) and the fact that we output $\theta^*$ if IAT identifies no accurate hypothesis. For the privacy guarantee, first note that any prefix of the noisy hypotheses $\theta^1, \ldots, \theta^t$ satisfies $\varepsilon_t$-differential privacy because of our instantiation of the Laplace mechanism (see the full version for the $\ell_1$ sensitivity analysis) and noise-reduction method NR. Then the ex-post privacy guarantee directly follows Lemma 2.8.   □

## 4   Experiments

To evaluate the methods described above, we conducted empirical evaluations in two settings. We used ridge regression to predict (log) popularity of posts on Twitter in the dataset of [1], with $p = 77$ features and subsampled to $n = $100,000 data points. Logistic regression was applied to classifying

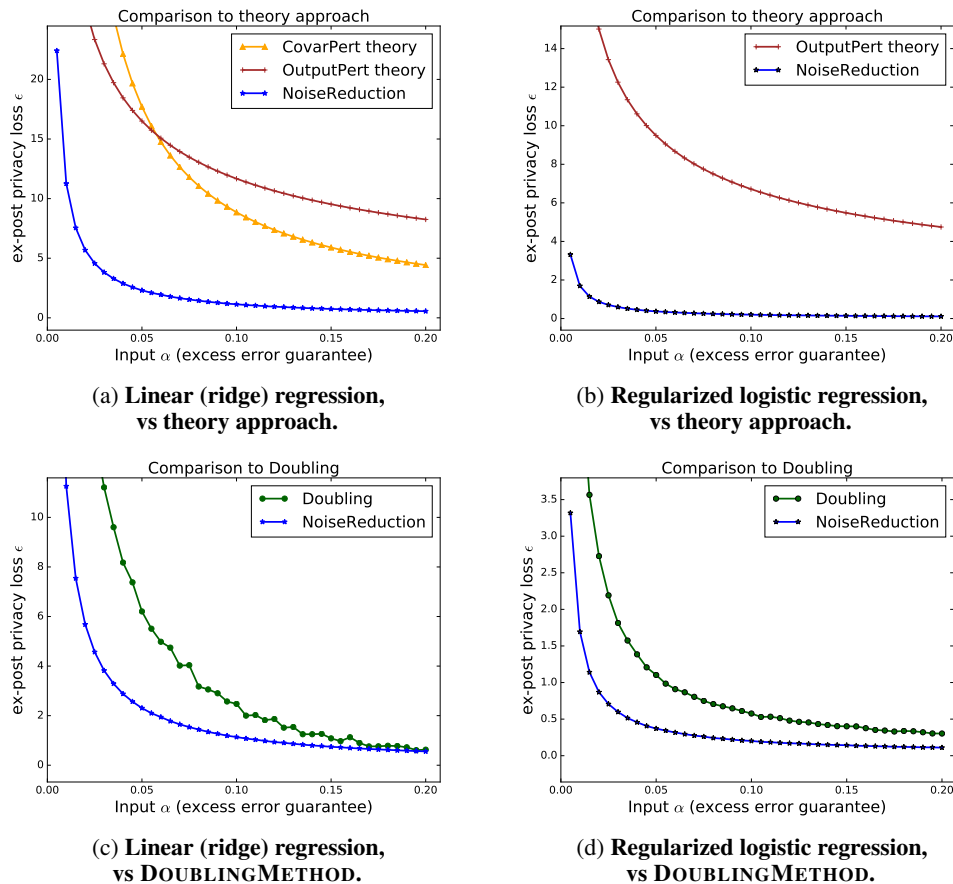

(a) **Linear (ridge) regression,
vs theory approach.**

(b) **Regularized logistic regression,
vs theory approach.**

(c) **Linear (ridge) regression,
vs DOUBLINGMETHOD.**

(d) **Regularized logistic regression,
vs DOUBLINGMETHOD.**

Figure 1: **Ex-post privacy loss.** (1a) and (1c), left, represent ridge regression on the Twitter dataset, where Noise Reduction and DOUBLINGMETHOD both use Covariance Perturbation. (1b) and (1d), right, represent logistic regression on the KDD-99 Cup dataset, where both Noise Reduction and DOUBLINGMETHOD use Output Perturbation. The top plots compare Noise Reduction to the "theory approach": running the algorithm once using the value of $\varepsilon$ that guarantees the desired expected error via a utility theorem. The bottom compares to the DOUBLINGMETHOD baseline. Note the top plots are generous to the theory approach: the theory curves promise only expected error, whereas Noise Reduction promises a high probability guarantee. Each point is an average of 80 trials (Twitter dataset) or 40 trials (KDD-99 dataset).

network events as innocent or malicious in the KDD-99 Cup dataset [13], with 38 features and subsampled to 100,000 points. Details of parameters and methods appear in the full version.[6]

In each case, we tested the algorithm's average ex-post privacy loss for a range of input accuracy goals $\alpha$, fixing a modest failure probability $\gamma = 0.1$ (and we observed that excess risks were concentrated well below $\alpha/2$, suggesting a pessimistic analysis). The results show our meta-method gives a large improvement over the "theory" approach of simply inverting utility theorems for private ERM algorithms. (In fact, the utility theorem for the popular private stochastic gradient descent algorithm does not even give meaningful guarantees for the ranges of parameters tested; one would need an order of magnitude more data points, and even then the privacy losses are enormous, perhaps due to loose constants in the analysis.)

To gauge the more modest improvement over DOUBLINGMETHOD, note that the variation in the privacy risk factor $e^{\varepsilon}$ can still be very large; for instance, in the ridge regression setting of $\alpha = 0.05$,

Noise Reduction has $e^\varepsilon \approx 10.0$ while DOUBLINGMETHOD has $e^\varepsilon \approx 495$; at $\alpha = 0.075$, the privacy risk factors are $4.65$ and $56.6$ respectively.

Interestingly, for our meta-method, the contribution to privacy loss from "testing" hypotheses (the InteractiveAboveThreshold technique) was significantly larger than that from "generating" them (NoiseReduction). One place where the InteractiveAboveThreshold analysis is loose is in using a theoretical bound on the maximum norm of any hypothesis to compute the sensitivity of queries. The actual norms of hypotheses tested was significantly lower which, if taken as guidance to the practitioner in advance, would drastically improve the privacy guarantee of both adaptive methods.

## 5 Future Directions

Throughout this paper, we focus on $\varepsilon$-differential privacy, instead of the weaker $(\varepsilon, \delta)$-(approximate) differential privacy. Part of the reason is that an analogue of Lemma 2.8 does not seem to hold for $(\varepsilon, \delta)$-differentially private queries without further assumptions, as the necessity to union-bound over the $\delta$ "failure probability" that the privacy loss is bounded for each query can erase the ex-post gains. We leave obtaining similar results for approximate differential privacy as an open problem. More generally, we wish to extend our ex-post privacy framework to approximate differential privacy, or to the stronger notion of concentrated differential privacy [3]. Such results will allow us to obtain ex-post privacy guarantees for a much broader class of algorithms.

## Footnotes

[1] A full version of this paper appears on the arXiv preprint site: https://arxiv.org/abs/1705.10829.

[2]If $\mathcal{A}$'s output is from a continuous distribution rather than discrete, we abuse notation and write $\Pr[\mathcal{A}(D) = o]$ to mean the probability density at output $o$.

[3]In fact, there are many applications beyond our own in which the sequence of queries input to AboveThreshold might be the result of some private prior computation on the data, and where we would like to release both the stopping index of AboveThreshold and the "query object." (In our case, the query objects will be parameterized by learned hypotheses $\theta_1, \ldots, \theta_T$.)

[4]This result does not follow from a straightforward application of privacy odometers from [17], because the privacy analysis of algorithms like the noise reduction technique is not compositional.

[5]We study the ridge regression problem for concreteness. Our method works for any ERM problem with strongly convex loss functions.

[6] A full implementation of our algorithms appears at: https://github.com/steven7woo/Accuracy-First-Differential-Privacy.

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
