[Reviews · NeurIPS 2017]

Reviewer 1



Pros: 1. Natural and important question: in many practical scenarios achieving a certain level of accuracy is crucial while privacy is a secondary concern. 2. Simple and easy to implement meta-approach based on gradual release of private information technique (from Koufogiannis et al 2015) combined with AboveThreshold technique. 3. Examples of applications together with an experimental evaluation Cons: 1. Using gradually decreasing levels of privacy with AboveThreshold is a rather obvious approach to the problem so should be considered the baseline in this case. The addition of the gradual release technique is nice but limits strongly the kind of approaches one can use with it off the shelf. Most importantly, as pointed out by the authors, for now its unknown how to use it with the more powerful approximate differential privacy. I suspect that the improvement from this part is relatively small (the smallest value of epsilon is still likely to dominate) so it is not clear if it's useful. 2. A comparison of the above mentioned baseline used with stronger algorithms for the tasks described could would have been very interesting but unfortunately is not included. Overall the work addresses an important question and describes some interesting ideas and a bit of experimental evaluation. The main drawback is that perhaps the more novel part of this paper seems of limited use for now but I think the work would still be of interest to those working on learning with privacy. Update: I agree with the authors that the definition and application of ex-post privacy are interesting enough to merit acceptance.

Reviewer 2



The authors present a framework for carrying out privacy-preserving optimization achieving target accuracy level rather than achieving target privacy level as usually studied in the literature. The authors show that for perturbation(input or output) based methods, one can apply the gradual release algorithm in [14] to achieve better privacy bound compared to the commonly used "doubling trick". The authors apply the framework to the ridge and logisitc regression problems and presented theoretical bounds in each case. Experiments are carried to validate the approach. This paper addresses the important problem of privacy-preserving learning. It takes a new angle on considering the accuracy first. This can be useful in the case when the application requires some minimal model quality. The approach in the paper is a clever application of the method developed in [14]. The tool developed in the paper can be useful in many similar applications. However, the framework in the paper is limited to the pure privacy and to the perturbation based methods. It would be useful for the authors to discuss these limitations. In addition, it would be good to compare to the method of using AboveThreshold with differentially private model evaluation (which can usually be done with very high accuracy.)

Reviewer 3



Summary: This paper considers a different perspective on differentially private algorithms where the goal is to give the strongest privacy guarantee possible, subject to the a constraint on the acceptable accuracy. This is in contrast to the more common setting where the we wish to achieve the highest possible accuracy for a given minimum level of privacy. The paper introduces the idea of ex-post differential privacy which provides similar guarantees to differential privacy, but for which the bound on privacy loss is allowed to depend on the output of the algorithm. They then present a general method for private accuracy constrained empirical risk minimization which combines ideas similar to the noise reduction techniques of Koufogiannis et al. and the Above Threshold algorithm. At a high level, their algorithm computes ERM models for increasing privacy parameters (using the noise reduction techniques, rather than independent instantiations of a private learning algorithm) and then uses the above threshold algorithm to find the strongest privacy parameter satisfying the accuracy constraint. The noise reduction techniques allow them to pay only the privacy cost for the least private model they release (rather than the usual sum of privacy costs for the first k models, which would be required under the usual additive composition laws) and the modified above threshold algorithm has privacy cost scaling only logarithmically with the number of models tested. They apply this algorithm to logistic and ridge regression and demonstrate that it is able to provide significantly better ex-post privacy guarantees than two natural baselines: inverting the privacy privacy guarantees to determine the minimum value of epsilon that guarantees sufficiently high accuracy, and a doubling trick that repeatedly doubles the privacy parameter until one is found that satisfies the accuracy constraint. Comments: Providing algorithms with strong utility guarantees that also give privacy when possible is an important task and may be more likely to be adopted by companies/organizations that would like to preserve privacy but are unwilling to compromise on utility. The paper is clearly written and each step is well motivated, including discussion of why simpler baselines either do not work or would give worse guarantees.